# Recent Developments in Effective Antioxidants: The Structure and Antioxidant Properties

**DOI:** 10.3390/ma14081984

**Published:** 2021-04-15

**Authors:** Monika Parcheta, Renata Świsłocka, Sylwia Orzechowska, Monika Akimowicz, Renata Choińska, Włodzimierz Lewandowski

**Affiliations:** 1Department of Chemistry, Biology and Biotechnology, Bialystok University of Technology, Wiejska 45E, 15-351 Bialystok, Poland; m.parcheta@pb.edu.pl (M.P.); w.lewandowski@pb.edu.pl (W.L.); 2Solaris National Synchrotron Radiation Centre, Jagiellonian University, Czerwone Maki 98, 30-392 Krakow, Poland; sylwia.orzechowska@uj.edu.pl; 3M. Smoluchowski Institute of Physics, Jagiellonian University, Łojasiewicza 11, 30-348 Kraków, Poland; 4Prof. Waclaw Dabrowski Institute of Agriculture and Food Biotechnology–State Research Institute, Rakowiecka 36, 02-532 Warsaw, Poland; monika.akimowicz@ibprs.pl (M.A.); renata.choinska@ibprs.pl (R.C.)

**Keywords:** flavonoids, antioxidants, SPLET mechanism, DPPH, FRAP, synchrotron

## Abstract

Since the last few years, the growing interest in the use of natural and synthetic antioxidants as functional food ingredients and dietary supplements, is observed. The imbalance between the number of antioxidants and free radicals is the cause of oxidative damages of proteins, lipids, and DNA. The aim of the study was the review of recent developments in antioxidants. One of the crucial issues in food technology, medicine, and biotechnology is the excess free radicals reduction to obtain healthy food. The major problem is receiving more effective antioxidants. The study aimed to analyze the properties of efficient antioxidants and a better understanding of the molecular mechanism of antioxidant processes. Our researches and sparing literature data prove that the ligand antioxidant properties complexed by selected metals may significantly affect the free radical neutralization. According to our preliminary observation, this efficiency is improved mainly by the metals of high ion potential, e.g., Fe(III), Cr(III), Ln(III), Y(III). The complexes of delocalized electronic charge are better antioxidants. Experimental literature results of antioxidant assays, such as diphenylpicrylhydrazyl (DPPH) and ferric reducing activity power assay (FRAP), were compared to thermodynamic parameters obtained with computational methods. The mechanisms of free radicals creation were described based on the experimental literature data. Changes in HOMO energy distribution in phenolic acids with an increasing number of hydroxyl groups were observed. The antioxidant properties of flavonoids are strongly dependent on the hydroxyl group position and the catechol moiety. The number of methoxy groups in the phenolic acid molecules influences antioxidant activity. The use of synchrotron techniques in the antioxidants electronic structure analysis was proposed.

## 1. Introduction

Reactive species are byproducts of crucial cellular processes and include reactive oxygen species (ROS) and reactive nitrogen species (RNS). They are formed in the mitochondrial respiratory chain, in the metabolism of purine nucleotides and arachidonic acid, prostaglandin synthesis, and enzymatic reactions catalyzed by, among others, xanthine oxidase and tryptophan dioxygenase [1]. Reactive species are involved in numerous processes occurring in the organism, e.g., cell maturation, immune protection, cytotoxicity against pathogens. However, they may cause damage to proteins, lipids, and DNA [2]. Under physiological conditions, excessively produced reactive species are neutralized by antioxidants, which can attenuate their damaging effects [3]. These include reactive species scavenging molecules such as superoxide dismutase and catalase, enzymes that repair oxidative damage, and methionine sulfoxide reductase [4]. In this case, when the overproduction of reactive species exceeds the antioxidant capacity of cell defense mechanisms, a phenomenon termed oxidative stress occurs. The cause of such imbalance in the body can be xenobiotics, a decrease in the amount number of antioxidants needed, or increased production of ROS/RNS [5]. Oxidative stress and imbalance of antioxidant properties are often the cause of cancer or cardiovascular diseases [1]. It has been proven that natural antioxidants, such as ascorbic acid and tocopherols, protect against cancer and cardiovascular diseases. The most important natural antioxidants can be found in cereals, vegetables, fruits, oilseeds, legumes, cocoa products, beverages (tea, coffee, red wine, beer, fruit juices), herbs, and spices [6]. There is also a growing interest in the use of natural and synthetic antioxidants as functional food ingredients and dietary supplements [7]. Synthetic antioxidants, due to their accessibility and improved activity, find application not only in stabilizing fats, oils, and lipids in foods but also in the pharmaceutical industry and as preservatives in cosmetics [8].

Antioxidant properties are provided by polyphenols [9]. The occurrence of more than one hydroxyl group bonded to the benzene ring is a common feature of polyphenols [10]. Polyphenols are classified into various groups according to the differences in the number of phenol rings and differences in structural elements [11].

The most common antioxidant assays used in an analysis of food, nutrition, and supplements are free radical diphenylpicrylhydrazyl (DPPH), ferric reducing activity power assay (FRAP), and 2,2′-azinobis(3-ethylbenzothiazoline-6-sulfonate) (ABTS). DPPH antioxidant assay can be applied for the estimation of the antiradical activity of functional foods such as herbal extracts and natural or synthetic pure compounds due to high stability, experimental feasibility, and low cost of DPPH radical [12]. The DPPH method consists of the addition of potential antioxidants (AH) to the DPPH solution. In the reaction, the unpaired valence electron from the nitrogen atom in DPPH radical is reduced by the hydrogen atom from an antioxidant molecule. As a result, the DPPH-H hydrazine is formed [13]. The percentage of quenched DPPH radical can be analyzed using spectrophotometric measurements, as the decrease of absorbance at about 515–520 nm [14]. The antioxidant activity of the compound is expressed as the amount of antioxidants in the solution necessary for reducing the initial concentration of DPPH free radicals by 50% [15].

The second crucial antioxidant assay is FRAP, which evaluates the total antioxidant activity in the reaction of reduction of ferric tripyridyl triazine (Fe III TPTZ) complex to blue ferrous tripyridyl triazine (Fe(II)-TPTZ) form. The reaction has a place at low pH and in the presence of an antioxidant, which can be monitored by measuring the change in absorption at 593 nm [16]. The absorbance of antioxidants is compared to the absorbance of standard FeSO_4_ x 7H_2_O at the known scope of concentration.

ABTS^•+^ for the evaluation of antioxidant properties of natural compounds can be applied. Apart from good solubility in organic solvents, ABTS assay can be implemented over a wide range of pH values. The total antioxidant activity of the compound in the ABTS assay is designated using the spectrophotometric method [17]. The results of the ABTS assay are expressed as EC_50_ value, which corresponds to such a concentration of antioxidant that is required to decrease the initial concentration of ABTS radical cation by 50% [18]. The antioxidant properties are strongly related to the chemical structure of the compounds, i.e., the number of hydroxyl groups, their mutual position in the aromatic ring, and the degree of their esterification.

This paper summarized the antioxidant properties of enzymatic and nonenzymatic plant-derived compounds with particular emphasis on the effect of chemical structure on the antioxidant activity of the compounds. The literature experimental data have been compared with computational methods based on the theory of density functional (DFT) for the explanation of experimental results. The thermodynamic parameters have been allowed to indicate the reaction mechanisms responsible for antioxidant properties and to explain the influence of the position and type of the substituent on the aromatic ring on the antioxidant ability of the compound. The proposition of the research on biologically effective ligands of antioxidant potential with the use of synchrotron radiation has been presented.

The motivation of the study was to search for more effective antioxidants and analyze the molecular mechanism responsible for antioxidant processes. The work, presenting the latest world reports on antioxidants, relates to our researches conducted within the scientific grants. In our researches, the relation between the molecular structure and electronic charge distribution, and biological activity, was analyzed. Moreover, the improvement of ligands antioxidant properties by metal complexation was investigated [19,20,21,22]. Our preliminary data show that the ligands complexation with antioxidant properties with some metals increases their antioxidant properties even by up to 10 times [23,24,25,26]. The increase of antioxidant properties is achieved especially as a result of the complexation by metals of high ion potential (expressed as a ratio of the ion charge to the radius), for example, Fe(III), Cr(III), Ln(III), Y(III). These metals stabilize the ligand’s electronic system and delocalize the electronic charge distribution. This effect increases ligand antioxidant properties. Therefore, in the presented work, the literature reports were confronted with our data concerning the improvement of ligands antioxidant properties by the selected metal complexation. The electronic charge distribution was analyzed with the use of spectroscopic methods (FT-IR, FT-Raman, NMR), X-ray diffraction, and quantum calculations. The literature and our consideration indicate that the molecular structure, even very complicated molecules, and the electronic charge distribution can be characterized using synchrotron techniques (Nobel Prize in Chemistry in 2003 for Roderick MacKinnon, 2009 for V. Ramakrishnan, A. Yonath, and T. Steitz, 2012 for B. Kobilka and R. Lefkowitz) [27,28].

## 2. Reactive Oxygen Species (ROS) and Reactive Nitrogen Species (RNS)

Free radicals are molecular fragments or molecules which have one or more unpaired electrons in their valence shell. The free radicals are very reactive. Many of them are unstable, and therefore either attach electrons from other molecules or donate their unpaired electrons [29]. They can act as oxidants or reducing agents [30]. The half-life of free radicals is very short (milli-, micro-, or nanoseconds) [31]. Free radicals are formed as a result of single-electron redox reactions. During photolysis, homolysis, radiolysis, and sonolysis of chemical compounds, reactions of the addition of other free radicals to double bonds of molecules, β-elimination, and oxidation of organic substances [32].

Reactive oxygen species (ROS) include not only free radicals (for example superoxide anion radical O_2_^•^^−^, hydroxyl radical ^•^OH, hydroperoxide radical HO_2_^•^, peroxyl radical ROO^•^), but also oxidants, such as singlet oxygen ^1^O_2_, hydrogen peroxide H_2_O_2_, hypochlorous acid HOCl, as well as oxygen and iron complexes, e.g., the ferryl radical Fe = O^2+^ [32,33]. Reactive oxygen species (free oxygen radicals, ions, neutral molecules) are formed as a result of the reduction of an oxygen molecule. One-electron reduction of the oxygen molecule produces superoxide radicals which, undergoing reduction, forms hydrogen peroxide (Formula (3)). Hydroxyl radical is generated as a result of the reduction of hydrogen peroxide in the Fenton reaction (Formula (1)) and Haber–Weiss reaction (Formula (2)). Further reduction of the hydroxyl radical leads to the formation of a water molecule (Formula (3)) [34,35]. Reactive nitrogen species (RNS) include: nitrosyl anion NO^−^, nitrile cation NO_2_^+^, peroxynitrite ONOO^−^, nitric oxide ^•^NO, and nitrogen dioxide ^•^NO_2_ [33].
Fe^2+^ + H_2_O_2_ → Fe^3+^ + ^•^OH + ^−^OH(1)
O_2_^•^^−^ + H_2_O_2_ → O_2_ + ^•^OH + ^−^OH(2)
O_2_ → O_2_^•^^−^ → H_2_O_2_ → ^•^OH → H_2_O(3)

Reactive species are generated during aging, mental stress, inflammation, ischemia, infection, and cancer. However, the sources of exogenous reactive species are dietary factors (coffee, alcohol, additives, barbecued, fried and grilled food, hydrogenated vegetable oils, processed food containing large amounts of lipid peroxides), ultraviolet radiation, X-rays, ionizing radiation, toxins, and drugs (benzopyrene, carbon tetrachloride, bleomycin, nitrofurantoin, mitomycin C), water pollutants (trihalomethanes, chloroform), air pollutants (formaldehyde, toluene, ozone, benzene, chlorine), chemical solvents (pesticides, perfumes, cleaning products, paints), and other factors such as smoking of tobacco products and automobile exhaust fumes. After entering the human body, exogenous compounds are broken down into free radicals [29,33,36,37,38,39,40].

Reactive oxygen and nitrogen species play a double role [40]. When they occur in the human body in low or medium concentrations, they take part in the processes of proper cell maturation (growth, proliferation, differentiation, and apoptosis of abnormal cells). They regulate immune processes and protect the human body against pathogenic microorganisms [41]. They destroy cancer cells, detoxify xenobiotics by Cytochrome P450, participate in the production of a universal energy carrier, ATP in mitochondria, activation of nuclear transcription factors, and transmission of signals within the cell and between cells [31,36,40,42,43]. However, when the balance between the production of reactive species and their removal is disturbed in the human body, oxidative stress or nitrosative stress occurs, respectively [32,40]. Then in the body, there may be a greater amount of endogenous and exogenous substances that easily undergo autooxidation, decreased production, and thus a smaller amount of low molecular weight antioxidants and decreased production of antioxidant enzymes [44]. ROS and RNS in too high amounts in the body destroy carbohydrates, proteins, nucleotides, and even entire cell organelles. They cause changes in the structure of DNA, which lead to mutations of the genetic material, as well as lipid peroxidation, which affects mainly unsaturated lipids from the membrane, producing damage to the integrity of this [32]. Lipid peroxidation cause changes in membrane fluidity. The most sensitive to lipid peroxidation are neurons due to unfavorable surface area to volume ratio [45]. Reactive species react with carbohydrates and break glycosidic bonds of important chemical compounds found in the human body, such as hyaluronic acid. They are responsible for the destruction of sugar residues of glycoproteins and glycolipids [32,40].

Glycoproteins are molecules present on the surface of the lipid bilayer of cell membranes [46]. There are many different glycoproteins with variable functions in the human body. Glycoproteins take part in cell-cell recognition (bacterium-cell, virus-cell, hormone-cell interactions) and cross-linking cells and proteins (e.g., collagen) in order to increase the strength and stability of the tissue. They make it easier for organ systems to communicate with each other. Glycoproteins perform transport functions (transferrin). These molecules may be hormones such as thyroid-stimulating hormone (TSH), human chorionic gonadotropin (HCG), and follicle-stimulating hormone (FSH), which is involved in growth, development, puberty, and reproduction. Thanks to glycoproteins, it is possible to bind the sperm to the surface of the egg. Another hormone, erythropoietin (EPO) which is a cytokine secreted by the kidneys, stimulates the production of red blood cells in the bone marrow under conditions of hypoxia. Thrombin, prothrombin, and fibrinogen are needed for blood clotting. The glycoproteins on the surface of red blood cells define the blood group. People with blood group A have A oligosaccharide, those with groups B and AB have B oligosaccharide and both A and B oligosaccharide, respectively, while people with blood group 0 do not have A and B oligosaccharide, but H oligosaccharide precursor only. Mucins located in mucus protect sensitive epithelial surfaces, including digestive, respiratory, urinary, and reproductive tracts. Glycoproteins (immunoglobulins such as IgG) participate in the body’s immune response. Moreover, they are located on the surface of B and T cells, where they are responsible for binding antigens [46,47].

Glycolipids are biomolecules consisting usually of alcohol sphingosine and a fatty acid with one (cerebrosides) or more (gangliosides) monosaccharides attached. Cerebrosides are present in a large quantity in brain white matter, nerve myelin sheaths, and muscles. They are found in small amounts in the cell membranes of other tissues [48,49]. Cerebrosides are involved in intracellular communication, cell agglutination, cellular development, and cytotoxic/antitumor effects [50]. In turn, gangliosides are present in very large amounts in the brain, especially in the gray matter [51]. They are found both in cell plasma membranes but also in nuclear membranes [52]. These molecules account for 10–12% of the lipids contained in the neuronal membrane [51]. They take part in cellular recognition through their specific sialoglycan components, adhesion, protein regulation, and modulate cell signal transduction. Moreover, they play a very important role in modulating intracellular and intranuclear calcium homeostasis [52] and in neuronal and brain development [51].

Moreover, ROS and RNS may cause changes in the cellular receptor functions, which are related to neurotransmitter and hormonal responses, prostaglandin formation, and interleukin activities [53].

Proteins are very sensitive to the action of free radicals. Reactive species influence the functions of structural proteins as well as the activity of enzymes. They cause inactivation, denaturation, and cross-linking of proteins and enzymes which contain sulfur. ROS and RNS reactions with proteins lead to the formation of i.a. protein hydroperoxides. Reactive oxygen and nitrogen species cause fragmentation of polypeptide chains, nitration of aromatic amino acid residues, oxidation of amino acids such as cysteine, methionine, tryptophan, tyrosine and, phenylalanine, and hydroxylation of aromatic and aliphatic amino acids. Most oxidized proteins lose their activity and are quickly removed from the body; however, some oxidized proteins accumulate in the human body and contribute to aging and the development of various diseases [40,44,53,54].

DNA, both nuclear and mitochondrial, is very susceptible to damage caused by free radicals, especially the hydroxyl radical. Reactive species break phosphodiester bonds, oxidize deoxyribose, and modify nitrogen bases. The attack of the ^•^OH radical on the C4-C5 double bond of the pyrimidine causes the formation of oxidative products of damage to these nitrogen bases, such as uracil glycol, thymine glycol, 5-hydroxydeoxyuridine, hydantoin, 5-hydroxydeoxycitidine, and urea residue. In turn, the attack of the hydroxyl radical on the purine bases causes the formation of 8-hydroxydeoxyguanosine, formamidopyrimidines, and 8-hydroxydeoxyadenosine. Reactive species contribute to the creation of DNA-protein cross-linking. Defective components of the respiratory chain are formed as a result of mitochondrial DNA damage. The increased leakage of electrons contributes to further cell destruction [40,55].

Reactive oxygen and nitrogen species also modify unsaturated fatty acids. As a result of the removal of the hydrogen atom from the methylene group of the unsaturated fatty acid, a lipid radical that has an unpaired electron on the carbon atom (^•^CH) is formed. The attack of molecular oxygen on the lipid radical causes the formation of the lipid peroxyl radical (LOO^•^). In turn, the lipid peroxyl radical splits spontaneously into hydrocarbons (e.g., pentane, hexane) and aldehydes (to a large extent, malondialdehyde—MDA). The resulting aldehydes can modify the amino groups in the side chains of amino acids and the purine and pyrimidine bases in the nucleic acid. Moreover, peroxyl radicals can react with other lipids to detach their hydrogen atoms, leading to the formation of lipid hydroperoxides (LOOH) and further lipid oxidation. Peroxidation of lipids contained in biological membranes disturbs their fluidity and permeability and changes their transport properties. Lipid oxidation leads to the depolarization of the mitochondrial membrane and the disturbance of oxidative phosphorylation. This causes the process of cellular respiration to be modified. When cytochrome c is released from the mitochondria and caspases are activated, cell death begins [40,53,56]. A general scheme of lipid peroxidation is demonstrated in Figure 1.

ROS and RNS reduce the production of ATP, decrease the level of total glutathione, increase the ratio of glutathione in the oxidized to the reduced form, GSSG/GSH, increase the concentration of calcium in the cell cytoplasm, and increase the permeability of biological membranes and their depolarization [32]. Ultimately, reactive oxygen and nitrogen species contribute to cell death, which may occur through apoptosis or necrosis [55]. During apoptosis, the cell contents are not released outward into the intercellular space, unlike necrosis. The necrotic cell content may cause additional oxidative stress [57].

## 3. Defense Mechanisms against Oxidative and Nitrosative Stress

Oxidative and nitrosative stress contribute to the development of many chronic diseases, including hypertension, diabetes, atherosclerosis, disturbance of wound healing, neoplastic diseases, eye disorders, brain disorders, neurodegenerative diseases such as Alzheimer’s or Parkinson’s disease, autoimmune diseases, and aging. Redox homeostasis disorders are prevented by antioxidants, which can be produced in the human body or are supplied externally along with food [38,39,57].

Both nonenzymatic (metabolic antioxidants such as uric acid, reduced glutathione (GSH), bilirubin, transferrin, coenzyme Q10, lipoic acid, and nutrient antioxidants such as vitamins A, C, E, flavonoids, carotenoids, trace metals (manganese, zinc, selenium), omega-6 and omega-3 fatty acids), and enzymatic antioxidants (such as catalase (CAT), superoxide dismutase (SOD), and glutathione peroxidase (GPx)) prevent and repair damage caused by reactive oxygen and nitrogen species, reduce the risk of degenerative and neoplastic diseases, and strengthen the body’s immune system [32,38,39,40,43].

Antioxidants are molecules capable of counteracting and slowing down the oxidation reactions of substances that can form free radicals. They neutralize or divert ROS such as CAT, SOD, and GPx, interrupt free radical reactions, and destroy ROS such as vitamin E, C, flavonoids, glutathione, uric acid, and also bind/inactivate of metal ions (catechins, ferritin, and ceruloplasmin). Thanks to this, they protect cells against damage [38,39,40].

There are three mechanisms of the body’s defense against ROS and RNS. The first line of defense, involving enzymes such as CAT and SOD, is to prevent reactive oxygen and nitrogen species from reacting with compounds essential to cells. The second, attended by, e.g., glutathione and uric acid that causes the termination of free radical chain reactions, and the third involves reparation and/or removal damage caused by the interaction of ROS and RNS with biomolecules. It is played by enzymes with oxidoreductase activity, e.g., paraoxonase or thioredoxin [36,59].

### 3.1. Nonenzymatic Antioxidants

#### 3.1.1. Vitamins

Vitamin E is one of the natural antioxidants. The group of the biologically active substances, such as tocopherols and tocotrienols, should be understood as vitamin E [1]. α-Tocopherol exhibits the highest biological activity among all of the vitamin E constituents, and it is the most common vitamin in the human organism. Tocopherols inhibit lipids peroxidation, through scavenging of lipid peroxyl radicals before it enters in reaction with adjacent fatty acid chains or of cell membranes proteins [60]. One of the tocopherol derivatives is Trolox. In contrast to fat-soluble tocopherol, Trolox exhibits the capacity to dissolution in water, which enables entering into different cell structures, both by the hydrophilic and hydrophobic side of the cell membrane [61]. Trolox protects cells against the toxic effect of hydrogen peroxide, reducing the ability of cells to uptake H_2_O_2_ into their internal structure and activation of antioxidant enzymes (glutathione, peroxidase, catalase) [62]. Trolox can be used as a reagent in measuring the total antioxidant capacity of all antioxidant components in plasma or serum, expressed as Trolox equivalent [63].

Among the vitamins included in the antioxidants also ascorbic acid, known as vitamin C should be mentioned. Vitamin C is well soluble in water.

Ascorbic acid is a natural component of food, or it can be supplemented. In the human body, it plays an important role in participating in hydroxylation reactions, which are crucial in the synthesis of collagen and carnitine. The biochemical activity of vitamin C is a result of its chemical chelating and reducing properties. The antioxidant role of ascorbic acid relies on the regulation of collagen synthesis and participation in the synthesis of prostacyclin and nitrous oxygen. Thus, vitamin C can maintain proper vascular function and reduce the risk of atherogenesis. Ascorbic acid is a cofactor for enzymes, e.g., dopamine hydroxylase [64].

Ascorbic acid regenerates vitamin E after its reduction during the scavenging of free oxygen radicals. The interaction between radicals of vitamins E and C can take place not only in homogenous solutions but also in liposomal membranes [65]. It is highly probable that ascorbates participate in protection from lipid peroxidation by oxidation of reduced form of α-tocopherol [66]. The stability of ascorbic acid in pharmaceuticals increases with the elevation in the number of bound transition metal ions that are responsible for catalyzing of autooxidation of ascorbic acid molecule. Ligands with strong moieties capable of metal ions chelation are responsible for catalytic oxidation of ascorbic acid [67].

#### 3.1.2. Flavonoids

A group of significant antioxidants is flavonoids, obtained from plant cells. The main role of flavonoids in plants is to provide an attractive flower color for pollinators. In leaf cells, they play a protective role from UV radiation and pathogens. Additionally, flavonoids participate in the energy transfer and biochemical processes of cells, strengthen the activity of plant hormones and growth regulators, take part in the process of cellular respiration and photosynthesis. Flavonoids are classified according to their biosynthetic origin. Some flavonoid subgroups, e.g., chalcones, flavanones, and flavan-3,4-diols, are intermediates in biosynthesis as well as end products accumulating in plant tissues. Other flavonoids groups are only known as biosynthesis end products. These are anthocyanidins, proanthocyanidins, flavones, and flavonols [68]. The second criterion by which flavonoids are classified is a chemical structure [69] (Figure 2).

This criterion was devised depending on which carbon atom in the C ring was connected to the B ring. Flavonoids, in which the aromatic B ring was attached to the carbon in the third position of the C ring, are classified as isoflavonoids. Neoflavonoids include compounds in which the B ring is connected in the C4 position of the C ring. Flavonoids in which the B ring is connected to the carbon atom in the second position of the C ring are divided into subgroups that include flavones, flavonols, catechins, anthocyanins, and chalcones [70]. Compounds classified as flavonoids have anticancer properties and are highly effective in preventing cardiovascular and neurodegenerative diseases. The effectiveness of flavonoids in preventing the above diseases depends on the absorption extent into the cells and spread in the body’s tissues [71]. The antioxidant properties of the flavonoids increase with the appearance in their structure of a hydroxyl group at the third carbon atom in the heterocyclic ring C, catechol moiety, resulting from the combination of hydroxyl groups with the 3’ and 4’ B positions, a double bond between 2 and 3 carbon atoms 4-oxo moieties on the C ring [72]. The structures of selected flavonoids are presented in Figure 3.

A logical series of different flavonoids are presented in Table 1 with an increasing number of hydroxyl groups and corresponding experimental values of TEAC_ABTS_^•^, TEAC_DPPH_^•^ (Trolox equivalent antioxidant capacity), and VCEAC_ABTS_^•^ (vitamin C equivalent antioxidant capacity).

The presence of the -OH group in the C_3_ position and the catechol moiety in the molecule increases the ability to capture the peroxyazotin radical compared to ebselen—a well-known antioxidant nitrous oxide radical [76]. The highest ability of quercetin compared to the flavonoids listed in Table 1 to prevent oxidative damage is partly due to the presence of a free hydroxyl group on the third carbon atom, responsible for stabilizing the flavonoid radical [77]. The torsion angle of the B ring, relative to the entire molecule, has a strong effect on the possibility of scavenging free radicals. The flavanol molecules containing the 3-OH substituent are flat, while those without a substituent in this position are twisted. The planarity of the molecule increases electron dislocation and stabilizes the flavonoid radical [78]. Removal of the hydroxyl substituent at the C_3_ position reduces coplanarity, deteriorating the ability to remove free radicals [78]. The substitution of the hydroxyl group at the C_3_ position with a methyl or glycosyl group completely abolishes the antioxidant activity of quercetin for β-carotene in linoleic acid [76]. The hydroxyl groups attached to the B ring form a hydrogen bond by linking the B ring to the C heterocyclic ring and the A ring. Removal of the hydrogen bond reduces the torsion angle of the B ring and reduces electron delocalization in flavonoid molecules [77]. The presence of an intramolecular hydrogen bond between the C_3_ hydroxyl group and the 3 ‘, 4’-catechol system explains the strong antioxidant properties of flavan-3-ols and flavon-3-ols [78].

Antioxidant activity underlies many biological properties, e.g., antiviral, anti-inflammatory, antibacterial, antiatherosclerotic, and anticancer [79]. Monophenols have significantly lower antioxidant activity than polyphenols, but structural changes result from the introduction of a group that gives or receives electrons at various positions of the phenyl ring, promoting the antioxidant activity of these compounds [80,81]]. The substitution of functional groups in the -ortho and -para positions seems to be a more effective way to increase the antioxidant capacity of a compound than substitution in the -meta position [82]. The strongest antioxidants among phenolic acids include their derivatives, in which the aromatic ring is substituted with three hydroxyl groups, such as 3,4,5-trihydroxybenzoic acid (gallic acid). Phenolic acids with two hydroxyl groups attached to the ortho ring, such as dihydroxycinnamic acid, 3,4-dihydroxybenzoic acid, and 2,4-dihydroxy benzoic acid, also demonstrate significant antioxidant properties. Compounds with one hydroxyl group attached to an aromatic ring, such as 2-hydroxybenzoic acid, 4-hydroxybenzoic acid, and 4-hydroxyphenyl acetic acid, exhibit the lowest antioxidant activity against free radicals [83]. Phenolic acids containing the same number of hydroxyl groups attached to an aromatic ring do not differ significantly in antioxidant properties. 4-hydroxy-3-methoxy benzoic acid has a higher antioxidant activity than 3-hydroxy-4-methoxy benzoic acid, which means that the position of the hydroxyl group has a significant effect on the antioxidant properties of flavonoids. The antioxidant activity of phenolic acid molecules elevates with the increase of methoxy group number [84].

Chlorogenic acid (CGA) is another active plant-derived antioxidant and a secondary plant metabolite [85]. CGA may also be a byproduct of anaerobic breathing from shikimic acid. The antioxidant properties of the chlorogenic acid molecule are the consequence of the presence of five active hydroxyl groups and one carboxyl group. The hydroxyl groups attached to the aromatic ring react with free hydroxyl radicals and superoxide anions [86]. CGA may inhibit xanthine oxygenase activity, thereby enhancing the activity of antioxidant enzymes, leading to inhibition of free radical formation and lipid peroxidation [87]. The antibacterial properties of chlorogenic acid rely on the inhibition of the bacteria growth, in particular *Escherichia coli, Staphylococcus aureus, Aspergillus niger,* and *Bacillus subtilis*.

Among the abundance of molecules involved in the human body’s defense mechanisms against oxidative stress, glutathione (l-γ-glutamyl-l-cysteinyl glycine, GSH) plays a significant role [88]. GSH occurs in the human body cells at a concentration of (0.1–10) mM [89]. Its antioxidant properties result from the presence of a thiol group, which participates in the reversible process of disulfide bond formation, crucial in maintaining the stability of water-soluble proteins. The formation and disintegration of bonds between sulfur atoms largely depend on the availability of electron donors or acceptors, determining the redox potential of the reaction environment. Thus, the presence of reduced thiols in the intracellular or extracellular environment has an impact on the structure and function of proteins. The equilibrium reaction illustrating the formation of a disulfide bridge is presented in Formula (4).
2R-SH ↔ R-SS-R + 2e^−^ + 2H^+^(4)

In aerobic biological systems, oxidation is coupled with the reduction of molecular oxygen. Reductases that require the presence of NADPH or NADH as an electron source participate in the reversible reduction reaction [90]. Glutathione activity is catalyzed by the enzyme glutathione peroxidase (GPx). Glutathione reduces harmful peroxides, protecting membranes and other cell components from oxidative damage.

### 3.2. Enzymatic Antioxidants

Enzymatic antioxidants naturally occur in the human body [43,53]. SOD (superoxide: superoxide oxidoreductase, also known as superoxide dismutase EC 1.15.1.1) is an endogenous antioxidant metalloenzyme. It protects the human body against the harmful effects of peroxides. This enzyme is a catalyst for the superoxide radical anion (O_2_^•−^) dismutation reaction (Formula (5)). This reaction produces hydrogen peroxide as a byproduct which can be removed by catalase or glutathione peroxidase. Superoxide dismutase can be broken down by enzymes such as peroxidase or catalase [43,53,59,91].
O_2_^•−^ + O_2_^•−^ + 2H^+^ → H_2_O_2_ + O_2_(5)

SOD isoforms differ in the number of subunits, cofactors, metals present in the active site of the enzyme, as well as the amino acid composition [53]. In mammals, there are three forms of superoxide dismutases: cytoplasmic, containing copper and zinc in its active center (SOD1, Cu, ZnSOD), mitochondrial (SOD2, MnSOD), and extracellular (SOD3, Cu, ZnSOD) with a structure other than cytoplasmic dismutase [40,43,91].

Catalase (CAT, EC 1.11.1.6), a tetrameric porphyrin-containing enzyme, is composed of four tetrahedrally arranged subunits. There are three types of catalase: typical or monofunctional such as mammal-type catalases, bifunctional, which exhibit catalase and peroxidase activity, and pseudo catalase, which is not an enzyme, but non-polar bis manganese III-EDTA-(${\mathrm{HCO}}_{3}^{-}$)_2_ complex, which degrades H_2_O_2_ to O_2_ and H_2_O. The target monofunctional catalase molecules are small organic substrates. These enzymes (glycoproteins) mainly show catalase activity. They are divided into catalases with small (55–69 kDa) and large (75–84 kDa) subunits. They differ not only in the size of the subunits but also in the heme prosthetic group. There is heme b in the small subunits (e.g., bovine liver catalase (BLC), while in the large subunits, heme d (e.g., *E. coli* HPII). Monofunctional catalases work over a wide pH range (5–10) and are resistant to organic solvents. Catalase-peroxidases are isolated from bacteria and fungi. This type of catalase has subunits of approximately 80 kDa. Bifunctional catalases are not glycoproteins, and their activity is pH-dependent. In turn, pseudo catalases (non-haem manganese-containing catalases) contain only three enzymes from different species of bacteria. Human catalase is located in the cytoplasm and peroxisomes of cells (erythrocytes, liver, kidney, and central nervous system cells). This enzyme is responsible for the decomposition of hydrogen peroxide into water and molecular oxygen. This reaction is two-step (Formulas (6) and (7)) [32,35,43,53,59,92]. Catalase-Fe (III) + H_2_O_2_ → compound I(6)
Compound I + H_2_O_2_ → Catalase-Fe (III) + 2H_2_O + O_2_(7)

CAT can exhibit peroxidase activity when the concentration of hydrogen peroxide is low. Then, it not only removes hydrogen peroxide but also carries out oxidation reactions of organic compounds such as formic acid, ethanol, methanol, or phenol (Formula (8)) [35,43].
(8)$$\mathrm{ROOH}+{\mathrm{AH}}_{2}\;\to H_{2}O\;+\;\mathrm{ROH}\;+\;A$$

There are two forms of GPx: selenium-dependent (GPx, EC 1.11.1.9) and selenium-independent (glutathione-S-transferase, GST, EC 2.5.1.18). These enzymes have various catalytic mechanisms, a different number of subunits, and a distinct way of binding selenium in their active center [17]. GPx (EC 1.11.1.9) catalyzes the reduction reactions of fatty acid hydroperoxides (ROOH) (Formula (9)) and hydrogen peroxide (H_2_O_2_) (Formula (1)) with the use of a glutathione molecule. It is two-electron oxidation of glutathione resulting in the formation of a free glutathione thiol radical [40,53].
ROOH + 2GSH → ROH + GSSG (glutathione disulfide) + H_2_O (9)
H_2_O_2_ + 2GSH → GSSG (glutathione disulfide) + 2H_2_O (10)

At the catalytic center of the GPx, the enzyme is a selenocysteine residue, in which selenium undergoes the redox cycle. Selenol (ESeH), responsible for the hydrogen peroxide reduction and organic peroxides, is oxidized to selenium acid (ESeOH). The next step (ESeOH) reacts with reduced glutathione (GSH), thereby forming a selenyl sulfide adduct (ESeGH). GSH regenerates in the reaction with the formed adduct, forming oxidized glutathione (GSSG) [78]. The whole cycle is demonstrated in Figure 4.

There are four GPx isoenzymes in the human body: extracellular GPx3 or GPx-P, cytosolic and mitochondrial GPx (cGPx or GPx1), cytosolic GPx2 or GPx-G1, and the phospholipid hydroperoxide GPx or GPx4. Most human tissues contain GPx but the liver contains the greatest amounts of this enzyme since the liver is the detox machinery of the human body [53].

Glutathione reductase (GR) is a flavoprotein enzyme. It removes glutathione disulfide (GSSG), which is harmful to cellular enzymes as its causes, inter alia, oxidation of protein thiol groups. This enzyme oxidizes NADPH, thus restoring reduced glutathione [43]. The reduced form of glutathione takes part in the neutralization of hydrogen peroxide produced in excessive amounts under oxidative stress [40,91].

Thioredoxin reductase (TRX) is a thiol-disulfide oxidoreductase. This enzyme in humans, plants, and bacteria was found. All thioredoxin proteins have an active site sequence Cys-Gly-Pro-Cys, which is highly conserved and share a common structure (five b-sheets and four α-helices) [93]. Thioredoxins are reduced by NADPH-dependent thioredoxin reductases (dimeric flavoproteins). TRX is responsible for the proper functioning of the cell. It activates transcription factors, such as activator protein 1 and nuclear factor *k*B, and stimulates cell growth. These enzymes are involved in the repair of proteins by supplying electrons to methionine sulfoxide reductases. TRXs undergo the reactions [94] of Formulas (11) and (12).
TRX-S_2_ + NADPH + H^+^ → TRX-(SH)_2_ + NADP^+^(11)
TRX-(SH)_2_ + Protein-S_2_ ↔ TRX-S_2_ + Protein-SH_2_(12)

TRX reduces the oxidized form of thioredoxin peroxidase. There are two types of thioredoxin: TRX1 and TRX2. TRX1 occurs in the cytosol, and TRX2 in the mitochondria. TRX is expressed in vascular smooth muscle cells (VSMCs) of normal arteries and endothelial cells [94].

Thioredoxin peroxidases (PRXs) are a family of antioxidant enzymes that are involved in the neutralization of small amounts of peroxides such as H_2_O_2_ and alkyl hydroperoxides. For this purpose, these enzymes use conserved cysteine residue at the NH_2_-terminal region to oxidize H_2_O_2_. In mammals, there are six isoforms of peroxiredoxins. They are divided into three subgroups: PRXs I-IV (2-Cys), PRX V (atypical 2-Cys), and PRX VI (1-Cys). Peroxiredoxins are found in the mitochondria, nucleus, cytosol, peroxisomes, and endoplasmic reticulum (ER) [53].

Ceruloplasmin (CP) is a copper-containing glycoprotein responsible for the transport of copper to tissue sites and serum antioxidation. It also acts as an aromatic amine oxidase. It prevents antioxidant stress by scavenging free radicals and superoxide ions. CP is also a catalyst for the in vitro oxidation of low-density lipoproteins (LDL). Increased synthesis and secretion of ceruloplasmin are observed during infections, inflammations, and diseases such as cancer, diabetes, and cardiovascular disease [95,96].

Paraoxonases comprise a group of three enzymes: PON1, PON2, and PON3. Their role is to inactivate pro-inflammatory and pro-oxidant mediators, regulate cell proliferation, and the xenobiotics and drug metabolism. PON1 and PON3 are formed in the liver. Then they are secreted into circulation. There, they bind to HDL (high-density lipoproteins) particles. PON2, on the other hand, is an enzyme located in the mitochondria, in the endoplasmic reticulum membrane, and the perinuclear region. Paraoxonases have three activities: paraoxonase (organophosphates hydrolysis), lactonase (lactone hydrolysis), and arylesterase (aromatic esters hydrolysis). The best-known enzyme is paraoxonase 1 (PON1), which is a 45 kDa glycoprotein. It binds in the circulation to HDL particles, and to a lesser extent, to VLDL (very low-density lipoproteins) and chylomicrons particles. It protects LDL and HDL particles against oxidative damage (lipid peroxidation). Moreover, it inhibits the development of atherosclerosis [97].

Antioxidant enzymes are less versatile free radical scavengers than low molecular weight antioxidants. The reactions of reactive oxygen and nitrogen species with antioxidant enzymes are more specific [14].

## 4. The Relation Between the Ligand Structure and Its Antioxidant Properties

### 4.1. Phenolic Acids

Since the last few years, the development of computational methods based on the theory of density functional (DFT) finds wide usage, particularly in the analysis of experimental results, or even predicting the antioxidant properties of chemical compounds e.g., phenolic acids [98]. The use of the DFT method allows to describe the three crucial mechanisms involved in the free radical scavenging: hydrogen atom transfer (HAT), single-electron transfer—proton transfer (SET-PT), sequential-proton-loss-electron-transfer (SPLET) were found. In a one-step HAT reaction, the free radical is responsible for detaching the hydrogen atom from the antioxidant molecule.

The chemical compounds of high antioxidant capacity exhibit low binding dissociation enthalpy (BDE) values. O-H bonding exhibits a great tendency to dissociation and, in consequence, to interact with free radicals. Phenolic compounds (ArOH) undergo a hydrogen transfer reaction according to the reaction (13).
ArOH + ROO^•^ → ArO^•^+ ROOH (13)

The SET-PT mechanism relies on the two stages. The first stage consists of the creation of cation-radical ArOH^+•^ according to the reaction (14).
ArOH + ROO^•^ → ArOH^+•^ + ROO^−^(14)

Cation-radical in the second stage is deprotonated, forming the ArO^•^ radical under the reaction (15).
ArOH^+•^ → ArO^•^ + H^+^(15)

The proton resulting in the reaction (15) attaches to the ROO- anion. The reaction (14) and (15) are described by the adiabatic ionization potential (IP) and proton dissociation enthalpy (PDE). At low ionization potential values, the tendency to form a peroxide anion radical is high, resulting in antioxidant activity growth. The SPLET mechanism consists of proton detachment from the hydroxyl group of a phenolic compound (16). The next step is electron transfer from the phenoxide anion ArO^−^ to ROO^•^, forming the phenoxide radical (17).
ArOH → ArO^−^ + H^+^(16)
ArO^−^ + ROO^•^ → ArO^•^ + ROO^−^(17)

The stability of the reaction (16) describes the affinity of the phenoxide anion to the proton, proton affinity (PA), and the electron transfer process in the reaction (8) is described as electron transfer enthalpy (ETE). Therefore, the final products of these mechanisms are ArO^•^ and ROOH. Compounds exhibit good antioxidant properties if the new radical species created in the scavenging reaction are more stable and less harmful to the cells of the human body than the initial free radical [99]. The results of the ferric reducing antioxidant power (FRAP) method on antioxidant properties in aqueous solution and using the DPPH^•^ radical in ethanol solution were compared to the thermodynamic parameters (BDE, IP, PDE, PA, ETE). It showed the mechanism responsible for antioxidant properties and explained the influence of the position and type of the substituent in the aromatic ring on the antioxidant ability of the compound.

DPPH^•^ radical (2,2′-diphenyl-1-picrylhydrazyl) is a stable artificial free radical, used in the measurement of the free radical scavenging capacity of the phenolic compounds in ethanol and aqueous systems. FRAP method consists of reduction of Fe(III) to Fe(II) to determine the antioxidant capacity of phenolic acids [75]. The IC_50_ means the concentration of sample needed to cause 50% inhibition. The IC_50_ values for DPPH assay and FRAP assay results expressed as the amount of equivalent of Trolox (mol TE/mol phenolic acid) for the BDE, IP, PDE, PA, and ETE thermodynamic parameters are summarized in Table 2.

Enthalpy of O-H bond dissociation decreases with the increase in the number of introduced methoxy groups. It corresponds to the experimental increase in antioxidant properties. It is noticeable in the syringic acid example, whose BDE 4-OH value decreased by 10 kcal/mol in comparison to 4-hydroxybenzoic acid. Reduction of the BDE 4-OH protocatechuic acid value by 7.6 kcal/mol relative to BDE 4-OH acid 4-hydroxybenzoate in ethanol indicates the elevation in antioxidant capacity along with an increase in the number of hydroxyl groups in the aromatic ring of a chemical compound [75]. SPLET is the primary mechanism in the FRAP methodology because the ETE values, in most cases, coincide with the experimental data.

It is also noteworthy that BDE values for 3-OH protocatechuic acid are greater than BDE 4-OH. It suggests that this acid has a greater tendency to dissociate the proton from the 4-OH substituent. It can be explained by the formation of a weak O3H …O4 hydrogen intramolecular bond [100]. The bond strength calculated by the NBO method for gallic acid proves that the electron and proton are removed from the 3-OH position because the bond in the hydroxyl group in this position is the weakest [101]. Enthalpy IP and PDE, describing the SET-PT mechanism, are dependent on the polarity of the solvent [102]. The IP value depends on the structure of the molecule and relocation [103]. The lower the IP value, the easier the compound dissociates protons. The introduction of hydroxyl or methoxy groups reduces the ionic potential values. The comparison of experimental results to calculated data shows that the HAT, SET-PT, and SPLET mechanisms describe the scavenging of DPPH^•^ free radicals. The lower the HOMO energy value is, the weaker the ability of the molecule to give electrons. When the HOMO energy value is higher, the better electron donor is the molecule. Because the deprotonation reaction involves electron transfer, HOMO’s energy distribution identifies the molecule structure responsible for antiradical activity [104]. The antioxidant capacity of chemical compounds, including phenolic acids, is also affected by spin delocalization. For example, it depends on the attachment method of the carboxyl group to the aromatic ring. Low antioxidant activity of benzoate radicals in comparison to radical homologs of cinnamic acid results from a smaller spin relocation of the COOH group compared to the -CH = CHCOOH group. The spin density of the carboxyl group is insignificant, and most of the spins are located in the ring or within the oxygen atoms of the substituent groups [102,105].

### 4.2. Flavonoids

The ability of flavonoids to scavenge free radicals is associated mainly with the presence of hydroxyl groups at the core position. Comparison of BDE, IP, PDE, PA, and ETE enthalpies allows predicting the preferred active site in the antioxidant structure, responsible for the inactivation of free radicals [102]. The minimum BDE value of the O-H bond shows which of the hydroxyl group, present in the flavonoid core, has the hydrogen atom most susceptible to dissociation. Most of the OH groups responsible for antioxidant activity are attributed to hydroxyl groups which are associated with the catechol moiety of the B ring and 3-OH of the C ring (Figure 2) [106]. In the case of quercetin, the minimum energy required to break the -OH bond refers to the 4’-OH moiety. Thus, if the HAT is the preferred reaction mechanism, then the 4’-OH is likely to participate in radical inactivation [107]. The total enthalpies value describing the individual mechanisms of free radical scavenging reactions for quercetin reach the lowest values in the 4’-OH position. It confirms that the hydroxyl group is significantly responsible for the antiradical activity. In Table 3, the values of total enthalpies describing the mechanisms of HAT, SET-PT, and SPLET reactions for quercetin were presented [108].

The reaction between free radical and flavonoid is presented in reaction (18).
R^•^ + FOH → RH + FO^•^,(18)
where R^•^ is free radical, FOH is a flavonoid, and FO^•^ a less reactive, stable free radical.

The reaction of flavonoids with free radicals is dynamic and proceeds according to the following reactions:RH ⇄ R^•^ + H^•^(19)
FOH ⇄ FO^•^ + H^•^(20)

According to the boundary theory of orbitals in DFT, the higher the HOMO energy value of a molecule, the greater its ability to lose an electron. In the case where the HOMO energy for FOH is higher than the LUMO energy for FO^•^, the direction of reaction (19) is favored, and FOH exhibits high antioxidant activity. The HOMO and LUMO energy values for the FOH and FO^•^ forms for selected flavonoids are presented in Table 4.

The HOMO energy values for FOH, presented in Table 4, are higher than for RH, while the LUMO energies for FO^•^ are higher than for R^•^. It proves that the reaction (19) is favored [109]. Studies on the antioxidant capacity of flavonoids consist of their activity assessment in terms of direct scavenging of free radicals in the electron-donor or proton-donor mechanisms [110]. However, it turns out that these are not the only mechanisms underlying the antioxidant compounds. Their antioxidative mechanism may result from the ability of flavonoids to chelate metal ions; thus, they can be inactive in the generation of free hydrogens [111]. Attempts to estimate the relationship between the structure and antioxidant properties of flavonoid chelates from Fe(II) have shown that 3-OH in the C ring and catechol system of the B ring is more important for chelation than 5-OH [112]. The catechol system of the B ring plays a significant role in Cu(II) binding [113]. In the binding of Fe(II) ions, vicinal groups -OH (3 ‘, 4’, or 7, 8) deserve special attention. They perform equally important functions in Fe(II) ion binding as hydroxyl groups at C5 and C3 in conjugation to the ketone group at the C4 position [112].

Stoichiometry 1:2 complexation reaction is preferred compared to 1:1, 2:2, and 2:3 stoichiometry [114]. The formation of the active redox Fe^3+^ complex—flavonoid is the initial stage in the scavenging of peroxide radicals [115]. The comparison of free energy of quercetin, catechin, and luteolin radicals creation, as well as the distribution of their spin density, allows determining the impact of individual OH groups on the antioxidant activity of these compounds.

Detachment of the hydrogen atom from the oxygen of the hydroxyl group is described by the difference between the heat of flavonoid formation and the corresponding radical formation (ΔHf). The ΔHf values for selected flavonoids are presented in Table 5 [116].

The data in Table 5 confirm that in the quercetin and kaempferol cases, the 3-OH group is the most involved in the radical formation. In the formation of the radical structure of (+)-catechin, luteolin and apigenin 4’-OH groups are involved the most.

## 5. Synchrotron Radiation in the Study of the Molecular Structure and Antioxidant Properties

Synchrotron techniques are an attractive and progressive tool that is leading to new advances in the fundamental understanding of materials structure. The power and controllability of the X-rays from the synchrotron have made possible a resolved structures with unprecedented precision. Photoemission experiments have provided much information about the electronic structure of solids, which is crucial to understanding the physical aspects of the structure of the material.

The knowledge of the electronic structure of biologically effective ligands and their complexes with metals is essential to understand their reactivity as well as their antioxidant properties. Nowadays, several synchrotron-based techniques are accessible to analyze the electronic and magnetic structure of these compounds. For example, photoelectron spectroscopy (PES) enables the calculation of experimental binding energies of valence or core electrons. In X-ray absorption spectroscopy (XAS), particularly in X-ray absorption near-edge structure (XANES) spectroscopy, core electrons are promoted to unoccupied electronic levels. XANES supplies structural and electronic data of the absorbing elements because of their sensitivity to the local bonding environment, such as the number of valence electrons, their spin configurations, the formal charge of the absorber, and the symmetry and coordination number. Therefore it is sensitive to the chemical nature and geometrical arrangement of the ligands [117].

Identification of the electronic structure of ligands complexed by transition metal ions is a fundamental problem in bioinorganic chemistry. The degree of protonation of the ligand is pivotal for hydrolysis reactions and hydrolytic enzyme action. In oxidative reactions, the question is whether reactive oxo-species (metal = O) exist or whether there is an oxo radical or water or a hydroxy group. The extended X-ray absorption fine structure (EXAFS) can supply valuable data on bond lengths however is usually insensitive to H atoms. The authors of the study [117] selected three coordination complexes [LMn^III^(acac)N_3_]BPh_4_, [LMn^III^(B_2_O_3_Ph_2_)(ClO_4_)], and [LMn^V^(acac)N] BPh_4_) that permitted to analyze ligand substitution and oxidation state change of an Mn ion with similar metal ligands (Figure 5).

The electron orbitals of ligands in 3d transition metal complexes can be analyzed by means of X-ray emission spectroscopy (XES). The authors carried out a comprehensive experimental and theoretical study of the X-ray emission from the valence band of Mn coordination complexes. The spectra show a signature of the ligand type and even allow a calculation of the ligand protonation degree. The Kohn–Sham orbitals of the density functional calculations help to understand the derivation of changes in spectra based on analysis of molecular orbitals that contribute to the spectral features [117].

In the paper [118], the authors report a combined solid-state NMR/X-ray powder diffraction (SSNMR/XRD) analysis for catechin. Differences in the powder diffraction patterns for all catechin forms are known, but further crystallographic information was unavailable. The full crystal structure for the tetrahydrate using SSNMR and synchrotron XRD with a high-resolution multianalyzer diffractometer has been discussed. The crystal structures supply significantly more insight than just atomic positions: for example, optical properties, magnetic susceptibility, and electrical conductivity can be predicted. Synchrotron XRD data afford all heavy-atom positions in (+)-catechin 4.5-hydrate and establish the space group as C2. SSNMR data (13C tensor and 1H/13C correlation) complete the conformation by providing catechin’s five -OH orientations. Positions of OH in C-H bonds are usually deduced from geometry [119].

The work [120] describes how the combination of resonant and non-resonant X-ray emission spectroscopies completed by model calculations allows for quantitative analysis of electronic states in 5d transition metal and metal-oxide materials. The use of X-rays supplies element selectivity that, in combination with the hard X-rays, enables the determination of the electronic state’s composition. The functional properties of transition-metal-based materials are susceptible to the electronic band structure of the central transition metal. The elemental composition and the presence of defects or lattice distortions have a strong impact on these properties [120].

The electronic structure of transition metal elements, especially orbital contribution and the relative energy position of the highest occupied and the lowest unoccupied states, is of significant importance for the design of efficient antioxidants. According to [120], by tuning the incident energy around the absorption edge of the element, the unoccupied states are probed by electron excitation to the intermediate atomic state. At the same time, the information about the occupied electronic levels can be received by the X-ray photons detection released during the breakup of the intermediate state to the final atomic state. So, RXES (resonant X-ray emission spectroscopy) combines XAS and XES, enabling for precise investigation of the matter electronic structure, with element selectivity [121]. The research of Wach et al. demonstrated the great possibilities of a coupling of XAS/XES spectroscopies to map the around-Fermi electronic structure of 5d elements. One of the most important concepts in metals is the Fermi surface. The importance of the Fermi surface in metals is according to the fact that all electronic transport phenomena are due to the electrons on or near the Fermi surface. For example, the electric and thermal conductivities depend on the Fermi surface [122]. It is worth emphasizing that the routine-used electron-based methods, such as X-ray photoelectron spectroscopy (XPS), are often unable to specify the metal electronic states because they cannot be conducted under working conditions [120]. The X-ray emission and absorption spectra show initial and final electronic state distributions that can be modified by the effects of core-hole screening, natural lifetime broadenings of initial and final states, and transition probabilities [120,123]. In opposite to data given by XAS/XES technique, XPS can provide only the knowledge about the occupied electronic final state [123].

The synchrotron radiation-based Fourier Transform Infrared Microspectroscopy (FTIRM) has been recently used to determine oxidative stress, e.g., photo-oxidative damage, results of radiation in various types of cells such as fibroblast, carcinogenic, and plasmatic cells, being able to correlate the infrared signals shifts with lipid and protein peroxidation [124]. Enterocytes are a suitable model to analyze the mechanisms by which polyphenols exhibit their antioxidant potential. They are cells that as a first interact with polyphenols during the absorption process, hence they are exposed to the highest concentrations of polyphenols [124]. The knowledge of what occurs in these cells under oxidative stress and in the presence of polyphenols is insufficient. Berraza and Castilo report the protective role of three phenolic acids, gallic acid, caffeic acid, and chlorogenic acid, two flavonoid compounds, catechin and quercetin, one vanilloid, and capsaicin. Effects of oxidative stress may bring under the displacement/reduction of some infrared bands. The band ratios 1740 cm^−1^/2960 cm^−1^, 2920 cm^−1^/2960 cm^−1^, 3012 cm^−1^/2960 cm^−1^, and 1630 cm^−1^/1650 cm^−1^ for lipid oxidation, lipid saturation, lipid desaturation, and proteins aggregation have been analyzed, respectively [124]. The results of Bezzeza and colleagues suggested that analysis of FTIR and FTIRM spectra can supply valuable information on the oxidant and antioxidant compounds’ impact in cellular systems because in a single and fast analysis, it is possible to determine their effects on lipid and protein oxidation. The mentioned effects correlate with those observed by standard biochemical assays.

The DEMETER dual electron microscopy and spectroscopy beamline with the X-PEEM method of the Solaris National Synchrotron Radiation Centre in Poland, designed for complete electronic and magnetic structure determination using a photoemission microscope with soft X-rays as an excitation source, XAS, XMCD, and XPS spectroscopies, can provide detailed information on ligands complexed with metals to understand the molecular mechanisms of correlation between their structure and antioxidant properties. The main advantage of the method is the ability to select the energy of excitations and the possibility of tuning to the characteristic edge of the absorption. The spectral resolution of the beamline is E/dE = 3 × 10^3^–1.5 × 10^4^.

The serious question is how the electronic charge distribution in ligands changes as the result of complexing with metals and how it influences the antioxidant properties of ligands. The study of the correlation between the molecular structure and antioxidant properties with the use of synchrotron radiation may provide great information unavailable using conventional research methods. The design of biologically effective ligands of antioxidant properties is crucial in the food industry, medicine, and many other areas; therefore, it requires further investigations using synchrotron radiation techniques.

## 6. Conclusions

The review of recent developments in effective antioxidants, in particular, the structure and antioxidant properties, confirms that the antioxidant properties of flavonoids increase due to the presence of the hydroxyl group in the C3 position and the catechol moiety. The presence of an intramolecular hydrogen bond between the C3 group and the 3’4’ catechol system is responsible for the strong antioxidant properties of flavon-3-ol and flavan-3-ol. Antioxidant activity elevates with the increase of the methoxy group number in the phenolic acid molecules. It is caused by decreasing enthalpy of -OH bond dissociation, which corresponds with the increase in antioxidant properties. To predict which of the hydroxyl groups in the molecule participates in the creation of the radical structure, the heat of formation values of individual radicals can be analyzed. In the case of kaempferol and quercetin, the 3-OH hydroxyl group is the most involved in the formation of radical structure. Among the major findings is that with the agreement on the literature research, the more -OH or -OCH_3_ groups in a ring, the better we observe antioxidant properties. However, we point out and present preliminary evidence that: The distribution of the electronic charge and the delocalization of electronic charge improve the antioxidant properties of ligands. Complexation of ligands with metals of high ionic potential (Fe (III), Ln (III) Y (III)) increases antioxidant properties. So the improvement of the antioxidant properties does not depend only on one factor (the number of OH and OCH_3_ groups), but also other very important effects. These properties were summarized in Table 1, Table 2, Table 3, Table 4 and Table 5. The delocalization of electronic charge as a result of ligand complexing with metals of high ionic potential can lead to increasing the antioxidant properties, according to our preliminary studies, even by up to 10 times. Thus the research on complexes of biologically important ligands (e.g., phenolic compounds) with metals is promising for developing new effective antioxidants, useful in the food industry, medicine, and other fields, and requires further investigations. The new solutions in the electronic structure researches with the use of synchrotron methods have been presented. Synchrotron techniques such as XAS, XES, PES, XANES could be very helpful to analyze the electronic structure of ligands complexed with metals to improve their antioxidant properties.

## Figures and Tables

**Figure 1 materials-14-01984-f001:**
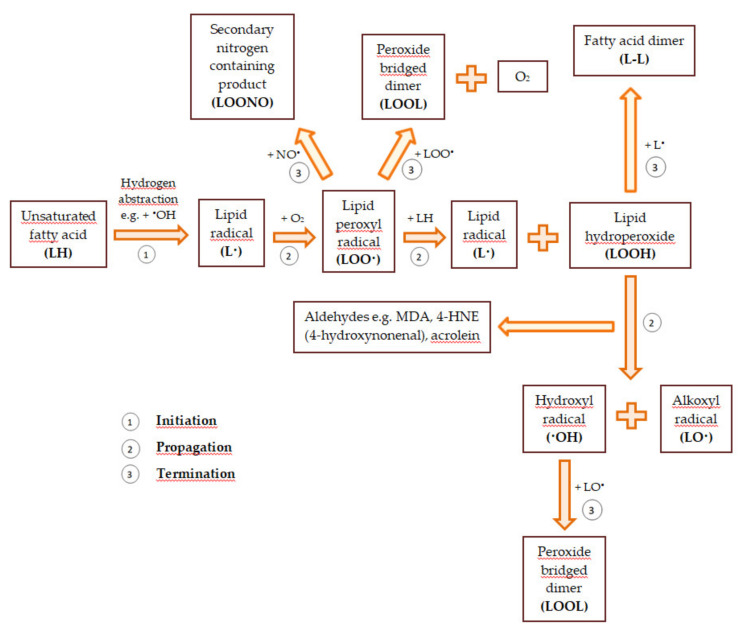
The general scheme of lipid peroxidation based on [57,58].

**Figure 2 materials-14-01984-f002:**
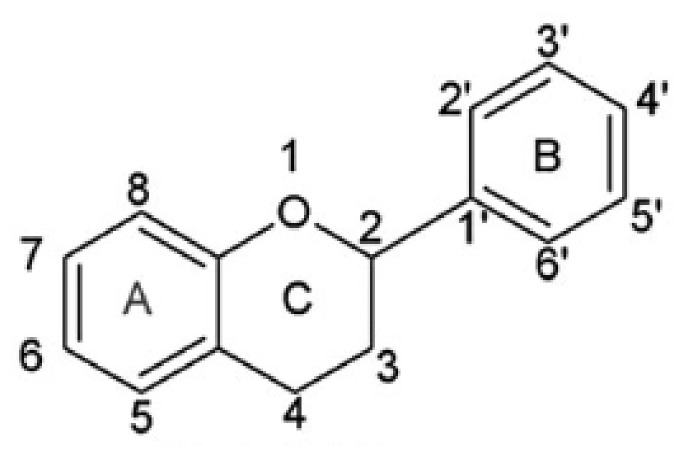
Skeleton of the initial structure of flavonoids.

**Figure 3 materials-14-01984-f003:**
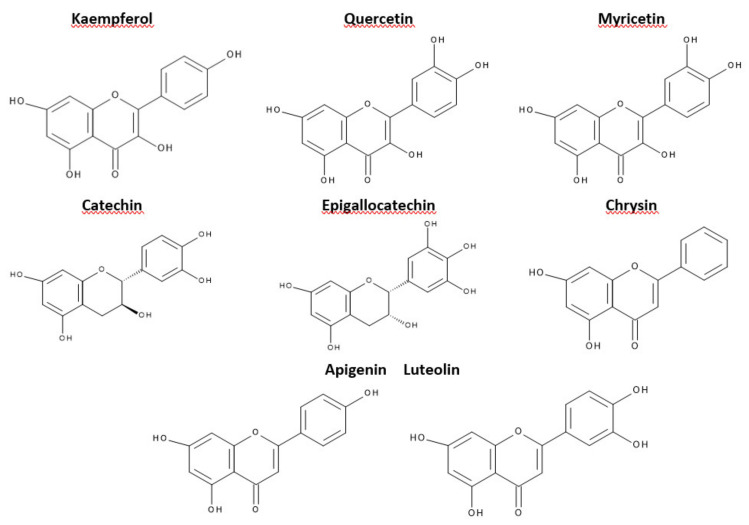
Structures of selected flavonoids.

**Figure 4 materials-14-01984-f004:**
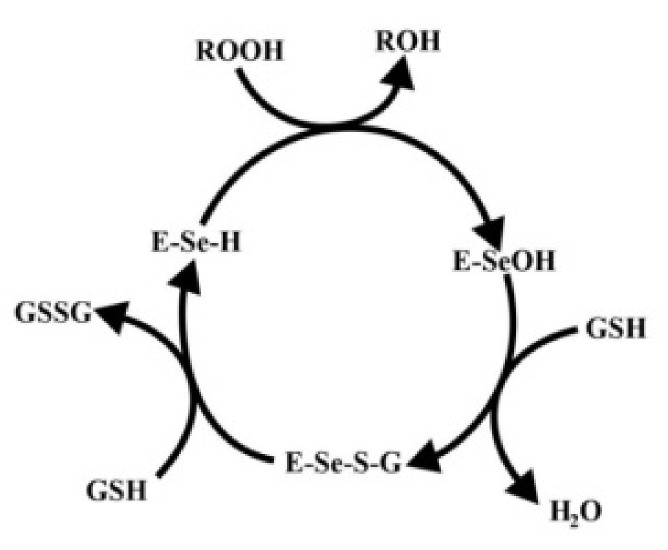
Catalytic mechanism of glutathione peroxidase. Based on [75].

**Figure 5 materials-14-01984-f005:**
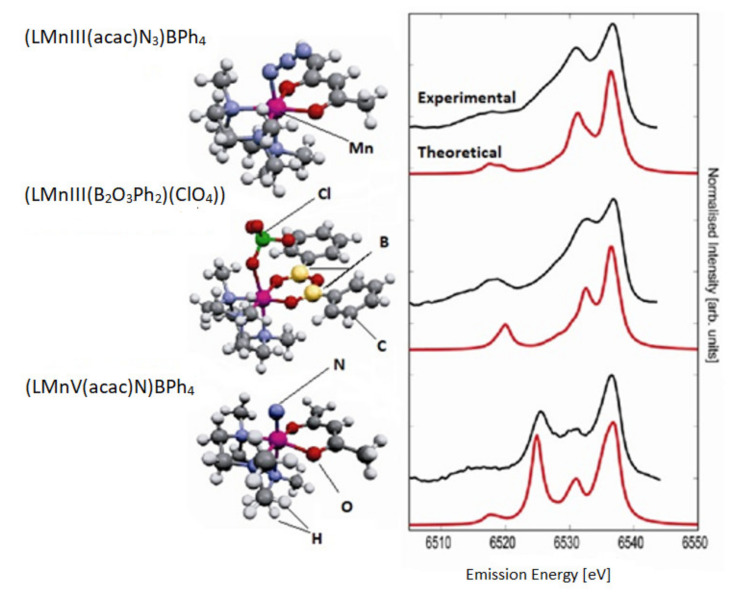
The experimental (top) and theoretical (bottom) valence-to-core XES spectra for three complexes. The shape of the experimental spectra is significantly dependent on the ligand type and oxidation state. Based on [117].

**Table 1 materials-14-01984-t001:** Flavonoids classified according to the increasing number of hydroxyl groups within the class together with the assignment of experimentally determined values of TEAC_ABTS•_, TEAC_DPPH_•, and VCEAC_ABTS_^•^. Based on [73,74,75].

Class of Flavonoid	Compound	TEAC _ABTS_^•+^ (mM) [73]	TEAC_ABTS_^•+^ (mM) [74]	TEAC_DPPH_ (mM) [73]	VCEAC_ABTS_^•^ (mg/L) [75]
Flavonol	Kaempferol	1.34	1.59 ± 0.017	1.32 ± 0.013	114.6 ± 3.3
	Quercetin	4.7	4.42 ± 0.081	4.60 ± 0.022	229.4 ± 3.8
	Myricetin	3.1	1.31 ± 0.012	1.38 ± 0.008	261.8 ± 2.9
Flavanol	Catechin	2.4	3.04 ± 0.030	2.95 ± 0.039	215.7 ± 2.6
	Epigallocatechin	-	3.71 ± 0.023	3.56 ± 0.018	264.4 ± 3.8
Flavon	Chrysin	1.43	0.081	0.053	24.9 ± 1.0
	Apigenin	1.45	0.086 ± 0.001	0.041	89.8 ± 5.6
	Luteolin	2.1	2.18 ± 0.015	2.24 ± 0.019	178.3 ± 2.3

**Table 2 materials-14-01984-t002:** IC_50_ and TEAC_FRAP_ values for phenolic acids with the calculated thermodynamic parameters: bond dissociation enthalpy (BDE), ionization potential (IP), proton dissociation enthalpy (PDE), proton affinity (PA), electron transfer enthalpy (ETE). Based on [75].

	4-Hydroxybenzoic Acid		Isovanillic Acid		Vanillic Acid		Protocatechuic Acid			Syringic Acid		
IC_DPPH_ (mM)	>1000		3.777 ± 0.160		3.38 ± 0.053		0.052 ± 0.002			0.043 ± 0.002		
TEAC_FRAP_ (mol TE/mol)	0.002 ± 0.001		0.155 ± 0.010		0.165 ± 0.019		1.46 ± 0.056			1.128 ± 0.037		
OH position	4 OH		3 OH		4 OH		3 OH		4 OH	4 OH		
Solvent	C_2_H_6_O	H_2_O	C_2_H_6_O	H_2_O	C_2_H_6_O	H_2_O	C_2_H_6_O	H_2_O	C_2_H_6_O	H_2_O	C_2_H_6_O	H_2_O
BDE (kcal/mol)	88.9	85.5	83.38	80.8	83.77	79.8	83.3	80.9	81.37	79.2	78.9	77.3
IP (kcal/mol)	129.2	120.6	118.8	110.9	118.9	110.8	120.5	112.6	-	-	115.8	108.3
PDE (kcal/mol)	6.2	13.3	11.0	17.0	1.31	9.8	9.31	5.5	7.3	13.7	9.71	6.1
PA (kcal/mol)	38.9	41.5	39.6	41.3	43.7	45.3	42.9	44.6	35.2	38.1	39.4	41.7
ETE (kcal/mol)	96.6	92.4	86.1	82.7	90.6	89.4	87.0	83.5	92.6	88.3	92.6	88.3

**Table 3 materials-14-01984-t003:** Reaction mechanisms enthalpies of HAT, SET-PT, and SPLET for quercetin. Based on [108].

Position of the Antiradical Active OH Group	HAT	SET-PT	SPLET
	BDE (kcal/mol)	IP + PDE (kcal/mol)	PA + ETE (kcal/mol)
3-OH	305	304	304
5-OH	373	372	372
7-OH	383	382	382
3’-OH	311	310	311
4’-OH	298	298	298

**Table 4 materials-14-01984-t004:** Flavonoids ordered by HOMO and LUMO energy values for the reduced form and the corresponding radical form. Based on [109].

Compound	FOH		FO^•^	
	HOMO (kcal/mol)	LUMO (kcal/mol)	HOMO (kcal/mol)	LUMO (kcal/mol)
Kaempferol	−127.573	−41.7921	−15.7003	44.7226
Quercetin	−126.192	−41.4784	−15.1795	44.6096
Myricectin	−125.941	−42.2314	−16.0078	44.1516
Chrysin	−138.366	−44.8042	−20.5258	43.3170
Apigenin	−135.793	−40.8509	−6.8148	39.4954

RH: HOMO = −153.9155, LUMO = 73.3057 R^•^ HOMO: −38.7110, LUMO: 16.6353.

**Table 5 materials-14-01984-t005:** Heat values of ΔHf formation of individual radicals for selected flavonoids. Luteolin and kaempferol do not create radicals in 3r and 3’r positions, respectively. Based on [116].

Flavonoid	ΔH_f_ (kcal/mol]				
	3r	5r	7r	3’r	4’r
(+)-Catechin	56.61	38.30	38.87	32.59	32.30
Luteolin	-	47.83	45.01	33.48	33.39
Kaempeferol	30.91	47.92	46.58	-	38.22
Qercetin	31.17	46.97	45.67	33.49	33.27
Apigenin	-	47.72	44.59	-	39.35

## Data Availability

Data is contained within the article.

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
