# Peer review of "Recent Developments in Effective Antioxidants: The Structure and Antioxidant Properties"

_materials, 2021, doi:10.3390/ma14081984_

Round 1
Reviewer 1 Report
The article materials-1159099, titled: Recent developments in effective antioxidants- the structure and antioxidant properties.
In this review, the authors present antioxidant compounds and analyze the parameters which influence antioxidant properties. They compare the antioxidant assays (DPPH and FRAP) with thermodynamic parameters obtained in silico. The different bond dissociation energies from OH groups of hydroxybenzoic acids and quercetin are compared and are discussed. Therefore, they describe the mechanisms of free radical creation. The antioxidant properties depend on the hydroxyl group position and catechol moiety in flavonoids, and the number of methoxy groups in the phenolic acids. Finally, the authors propose the use of synchrotron technique to analyze the antioxidant electronic structure.
In my opinion, this review is adequate to be published owe to the interesting research but with minor revision.
Minor revision:
- I would recommend to authors to make an abbreviations section and an index.
- In the third line of the introduction, the authors write: “in the metabolism of purine nucleotides, arachidonic acid, prostaglandin synthesis, and enzymatic reactions catalyzed by,”
I think it would be: “in the metabolism of purine nucleotides and arachidonic acid, prostaglandin synthesis,… ”
- In the phrase “the addition of potential antioxidant AH” The AH should be between parentheses. And in this paragraph should be referenced:
Sirivibulkovit, K.; Nouanthavong, S.; Sameenoi, Y. Paper-based DPPH Assay for Antioxidant Activity Analysis. Anal. Sci. 2018, 34, 795-800. This is the reference number 20.
- In the section 1, this following phrase is repeated before and after formulas 1,2 and 3: “Reactive nitrogen species (RNS) include: nitrosyl anion NO-, nitrile cation NO2+, peroxynitrite ONOO-, nitric oxide •NO and nitrogen dioxide •NO2 [30].”
- In the following phrase, the authors should say that the lipid peroxidation affects mainly unsaturated lipids from membrane producing a damage in the integrity of this. “They cause changes in the structure of DNA, which lead to mutations of the genetic material, as well as lipid peroxidation [42].”
- In this phrase “They are responsible for the destruction of sugar residues of glycoproteins and glycolipids [29, 37].” The authors should give some examples of glycoproteins and glycolipids and its importance. For instance, immunoglobulins, gonadotropic hormones, cytochrome transport proteins for glycoproteins and cerebrosides and gangliosides for glycolipids. The cerebrosides are important components in the cellular membrane of muscles and nervous cells while gangliosides constitute 6% membrane lipids from cells of grey material from brain.
- It is not necessary write (8-OHdG) because it is not renamed in all the paper.
- In the last phrase of the section 1. The capital letter of “The Necrotic cell” should be changed to “The necrotic cell”.
- In the Table 1, the width of the lines is not homogeneous.
- In the section 2.1.2. Flavonoids, page 10.
Among abundance of molecules, involved in the human body's defense mechanisms
against oxidative stress, glutathione (L-γ-glutamyl-L-cysteinyl glycine, GSH) plays a sig-
nificant role
(Fig. 7).
Correct it and write in the same line: nificant role (Fig. 7).
- In the section 2.2. Enzymatic antioxidants. When the authors talk about superoxide: superoxide oxidoreductase, they should say that this enzyme can be named superoxide dismutase because then in the text they name it as superoxide dismutase, and it can be confusing.
- Explain what pseudo catalase makes.
- Complete the following phrase:” Most human tissues contain GPx but the liver contains the greatest amounts of this enzyme”
since the liver is the detox machinery of the human body.
- Write what GR means glutathione reductase in the following phrase: GR is a flavoprotein enzyme.
- Write what TRX means thioredoxin reductase in the text.
- Change Cp to CP in the following phrase: Cp is also a catalyst for the in vitro oxidation…
- I do not understand this phrase: A characteristic feature of a compound with good antioxidant properties is the creation of radicals more stable and less harmful to the cells of the human body than the initial radical, which is a substrate in the reactions of one of the mechanisms presented above [98].
Author Response
Response letter to the Reviewer 1
We are very grateful to the Reviewer for valuable and detailed comments that help to improve this work making it more legible and useful emphasizing the information which can be a step to receiving more effective antioxidants and can provide complementary information for future experiments. A detailed description of the corrections is provided in file below.

Reviewer 2 Report
On one hand this review is pretty similar to some recent book chapters and reviews. The figures of the introductory part are similar, and it should also be mentioned that it is not rational which paragraph earns a figure or scheme. Please check the citations below:
https://www.ncbi.nlm.nih.gov/pmc/articles/PMC2763257/
https://www.intechopen.com/books/antioxidants/antioxidant-compounds-and-their-antioxidant-mechanism
https://www.sciencedirect.com/science/article/pii/S0223523419305276
https://www.intechopen.com/books/traditional-and-complementary-medicine/a-review-on-natural-antioxidants
https://www.mdpi.com/2076-3921/6/3/70
https://link.springer.com/article/10.1007/s00204-020-02689-3
On the other hand the presented investigation methods and possible conclusions based on those are interesting and unique. Therefore it is suggested to change the focus of the manuscript, give more exact explanations for the theoretical and experimental observations and emphasize more what can be concluded from these observations, what are the perspectives, which methods should be used for the design of better antioxidants etc.
Minor comments:
Fig 6. needs no numbering
Fig 2. Peroxide bridged dimer is not (.OH)
Few typos: Necrotic, IC50
Author Response
Response letter to the Reviewer 2
We are very grateful to the Reviewer for valuable and detailed comments that help to improve this work making it more legible and useful emphasizing the information which can be a step to receiving more effective antioxidants and can provide complementary information for future experiments. A detailed description of the corrections is provided in file below.

Reviewer 3 Report
The review manuscript related to the recent developments in effective antioxidants- the structure and antioxidant properties is the good effort made by authors. But I do not feel this script is suitable for this Journal. Anyway, its editor to decide the suitability of work to this Journal.
But still manuscript need thoroughly correction before it should be accepted for publication.
The main problem associated with this review script is that this script subject is not deeply touch. Before inquiring for the method of preparation of a review article, it is more logical to investigate the motivation behind writing the review article in question. The fundamental rationale of writing a review article is to make a readable synthesis of the best literature sources on an important research inquiry or a topic. For the specification of important questions to be answered, number of literature references to be consulted should be more or less determined. Discussions should be conducted with colleagues in the same area of interest, and time should be reserved for the solution of the problem(s)
Under the abstract, author have mention “Experimental literature results of antioxidant assays, such as DPPH and FRAP, were compared to thermodynamic parameters obtained with computational methods.” Always provide with full form of abbreviation used in the script.
Summarize the main findings, including the strength of evidence for each main outcome;
Provide a general interpretation of the results in the context of other evidence, and implications for future research
Research articles in the literature should be approached with a methodological, and critical attitude and finally data should be explained in an attractive way.
English of the script required correction from native speaker.
I suggest authors should follows the above points and revised the script carefully.
Author Response
Response letter to the Reviewer 3
We are very grateful to the Reviewer for valuable comments that help to improve this work making it more legible and useful emphasizing the information which can be a step to receiving more effective antioxidants and can provide complementary information for future experiments. A detailed description of the corrections is provided in file below

Round 2
Reviewer 2 Report
This reviewer appreciates the efforts of the authors for improving the manuscript significantly. Only one minor point appears now that probably too many figures have been removed, therefore the single flavonoid structure seems not enough to show the diversity of the flavonoid type antioxidants. Nevertheless, the manuscript can be accepted after adding some exact structures e.g. those that can be found in the tables.
Author Response
The letter in which authors refer to the comments is attached in the file below.

Reviewer 3 Report
The author have revised the script. But what are the question raised by reviewer are not provided under response to reviewer comments file. So how reviewer will understand that what are the question been asked. Kindly provide the first comment follow by response in order to evaluate the revised version of the script.
Author Response
The letter in which the authors refer to the comments is attached in the file below.
